# Knowledge, Attitudes, and Perception towards COVID-19 Vaccination among the Adult Population: A Cross-Sectional Study in Turkey

**DOI:** 10.3390/vaccines10020278

**Published:** 2022-02-11

**Authors:** Meliha Cagla Sonmezer, Taha Koray Sahin, Enes Erul, Furkan Sacit Ceylan, Muhammed Yusuf Hamurcu, Nihal Morova, Ipek Rudvan Al, Serhat Unal

**Affiliations:** 1Infectious Diseases and Clinical Microbiology Department, Hacettepe University Faculty of Medicine, Ankara 06100, Turkey; furkan.sacit@hacettepe.edu.tr (F.S.C.); muhammedhamurcu@hacettepe.edu.tr (M.Y.H.); nihal.morova@hacettepe.edu.tr (N.M.); ipekridvanal@hacettepe.edu.tr (I.R.A.); sunal@hacettepe.edu.tr (S.U.); 2Internal Medicine Department, Hacettepe University Faculty of Medicine, Ankara 06100, Turkey; takorsah@gmail.com (T.K.S.); eneserul@hacettepe.edu.tr (E.E.)

**Keywords:** vaccine, vaccination, vaccine hesitancy, health knowledge, health beliefs, knowledge, attitudes, perception

## Abstract

Background: The coronavirus disease 2019 (COVID-19) pandemic continues to wreak havoc on lives and ravage the world. Several vaccines have been approved for use against COVID-19; however, there may be hesitancy and negative perceptions towards vaccination, which may reduce the willingness to be vaccinated. Further, studies assessing the current perception toward COVID-19 vaccination are scarce. This study aimed to assess community knowledge, attitudes, and perceptions regarding COVID-19 vaccines among the general population of Turkey. Methods: A cross-sectional survey was carried out among 1009 adult participants from the 13–20 April 2021. Demographic data were collected, and attitudes and perceptions toward COVID-19 vaccines were evaluated. A multivariable regression analysis was performed to identify the factors predicting perception towards COVID-19 vaccinations. Results: Just over half of participants were male (52.6%) and the majority of respondents were aged between 30 and 39 years (33.8%). Our study revealed that 62.7% of participants had positive perceptions of COVID-19 vaccines. Logistic regression analysis results showed that older people (≥30 vs. <30) were less likely to have a positive perception towards COVID-19 vaccines (OR = 0.70, 95% CI = 0.51–0.94). We also found participants who had a previous history of influenza vaccines (OR = 2.01, 95% CI = 1.43–2.84), bachelor’s degrees or above (OR = 1.47, 95% CI = 1.12–1.91), and a personal history of COVID-19 (OR = 1.58, 95% CI = 1.10–2.26) were more likely to have a positive perception regarding COVID-19 vaccines. Conclusion: The proportion of the general population in Turkey who believe in COVID-19 vaccine effectiveness is not inferior to that of other countries. However, the low positive perception even among the population applying for vaccination indicates that understanding the perception of the general population and its influencing factors may contribute to developing a strategy for improving vaccination rates by addressing these factors.

## 1. Introduction

Towards the end of December 2019, an extraordinary outbreak of a pneumonia-like disease of unknown cause emerged in Wuhan, China [1]. The World Health Organization (WHO) subsequently termed the illness coronavirus disease 2019 (COVID-19); it is caused by a novel coronavirus called SARS-CoV-2 [2]. Despite quarantine efforts and strict global containment, COVID-19 has spread worldwide and poses a global health concern. As of January 2022, the COVID-19 outbreak has caused more than 350 million infections worldwide, as well as over 5.5 million deaths [3], after it was declared a global pandemic by the WHO on 11 March 2020 [4].

The SARS-CoV-2 pandemic has inflicted almost unimaginable harm on our lives in unprecedented ways. In this grim situation, the development of safe and effective vaccines against the virus is considered a decisive moment to curb disease spread and ensure the resumption of ordinary life. More than 330 candidate vaccines are currently under clinical evaluation globally, 44 of which are already in phase 3 clinical trials [5]. The three presently authorized COVID-19 vaccines were initially shown in trials to be highly effective in preventing the disease among adults, with an efficacy of 76.7% for the Ad26.COV2. S (Johnson & Johnson–Janssen, Beerse, Belgium) vaccine [6], an efficacy of 94.1% for the mRNA-1273 (Moderna, Cambridge, MA, USA) vaccine [7], and an efficacy of 95% for the BNT162b2 (Pfizer–BioNTech, Manhattan, NY, USA) vaccine [8]. In mid-September 2020, phase 3 studies on the inactivated SARS-CoV-2 vaccine named CoronaVac, developed by Sinovac Life Science Company (Beijing, China), were initiated in Turkey [9]. The Republic of Turkey Ministry of Health gave an emergency-use authorization for the CoronaVac COVID-19 vaccine on 14 January 2021. It launched an immediate vaccination program prioritizing healthcare workers and older adults aged 65 and over. After the limited number of doses of CoronaVac vaccine were provided and the high efficacy of the BNT162b2 mRNA COVID-19 vaccine was shown in clinical trials, the Republic of Turkey Ministry of Health launched a mass vaccination campaign by administering the BNT162b2 vaccine in April 2021. As of 30 July 2021, more than 26 million people, comprising almost 45% of the adult population, had received two doses of vaccine in Turkey [10]. Despite these groundbreaking scientific discoveries and mass vaccination campaigns, skepticism, hesitancy, and a negative perception of vaccines are obstacles to halting the outbreak. Concerns have also been raised about the effectiveness of the existing vaccines in protecting against new variants as new SARS-CoV-2 variants have emerged and spread. Although many vaccines have been found to be highly effective against the COVID-19 reference strain, they may not provide the same level of protection against mutation strains [11,12]. The omicron variant was first identified in South Africa in November 2021 and has quickly become the dominant variant after a third wave of COVID-19 driven by the delta variant had largely subsided [13]. A recent study found BNT162b2 vaccine effectiveness of 70% (95% CI: 62–76) during the omicron period [13].

Vaccine hesitancy, which encompasses reluctance, delay, or refusal to receive a vaccine despite availability, has been a significant threat to the effectiveness of vaccination programs [14]. There is a lot of concern and debate about the COVID-19 vaccine among the general population. Despite encouraging results for COVID-19 vaccine studies, according to a global survey of approval of COVID-19 vaccines, 48% of the study participants are unsure regarding COVID-19 vaccination and whether or not they will accept the vaccine [15]. In the study conducted simultaneously in Turkey and the UK, one out of three people in Turkey were unsure about receiving the COVID-19 vaccine, and 3% of the participants in both countries refused to be vaccinated [16]. Healthcare practitioner recommendations, prior influenza vaccinations, high perceived likelihood of getting COVID-19, and trust in government have been critical facilitators for vaccine acceptance [17,18].

Moreover, understanding the perception of the general population and their willingness to be vaccinated are crucial for improving vaccination rates. In this context, a recent study revealed that college and above educational level, access to mass media, and urban residency were significantly associated with awareness about COVID-19 vaccines [19]. Furthermore, the WHO strongly encourages governments to provide accurate and reliable knowledge about COVID-19 vaccination [20]. To undertake Turkey’s most effective vaccination strategy, we need to know the Turkish people’s attitudes, knowledge, and perceptions regarding COVID-19 vaccines. There is no previous study investigating knowledge, attitudes, and perceptions regarding COVID-19 vaccination among the general population in Turkey. Therefore, the present study aimed to assess the overall perception and identify factors predictive of positive perception towards COVID-19 vaccination in the adult population.

## 2. Materials and Methods

### 2.1. Study Design and Participant

This cross-sectional study was conducted from 13 April 2021 to 20 April 2021, among the general population of Turkey. Adult participants who applied to the Hacettepe University Hospital to receive the first dose of mRNA BNT162b2 Covid vaccine as part of a vaccination campaign were invited to answer the questionnaire. Respondents who understood the Turkish language, were 18 years old and above, and could give informed consent were recruited for the study. Healthcare professionals and people over 60 were relatively few in this study, as the Coronavac vaccine had already been administered.

The questionnaire begins with the study investigators’ contact information to ease communication between participants, an introduction section about the aim of the study and researchers, and a consent section in which participants could agree or refuse to participate in the study. The second part of the study was designed to collect sociodemographic data and general information from participants. It includes gender, education level, age, employment status, presence of chronic diseases, medication use, whether flu shots were received in the past, previous infection of family members or participants with COVID-19, and COVID-19 vaccine-related information sources. The last part was created to primarily emphasize the perception, knowledge, and beliefs surrounding COVID-19 vaccination efficacy. These were assessed based on a survey tool and guidance [21]. Some of the statements we used in this study were modified from previous studies on COVID-19 vaccination [22,23,24]. This section started with a question about whether the vaccines would produce an immune response against COVID-19. Based on this question, we divided our respondents into two main groups: those with a positive perception of vaccine efficacy and those with a negative perception of vaccine efficacy. These two groups were then compared in terms of gender, age, educational background, working status, the presence of chronic diseases, the prior history of participants with COVID-19, and a previous influenza vaccination history. This subsection also assessed the participants’ general perception and knowledge about COVID-19 vaccines through questioning eight statements as presented in Table 1 with a category (“agree,” “undecided,” and “disagree”). An English translation of the questionnaire is available in Appendix A at the end of the present paper.

### 2.2. Ethical Approval

The protocol was conducted in agreement with the Helsinki Declaration. We obtained informed consent from all participants before their participation. The ethics committee approval was obtained from the Hacettepe University Observational Research Ethics Committee (Ethics Committee Approval No:2021/13-59).

### 2.3. Statistical Analysis

The data were analyzed using IBM SPSS v. 24 (IBM Corp., Armonk, NY, USA). Categorical variables were represented as counts and percentages. The comparison of perception and knowledge between groups within the baseline characteristics was evaluated using the Fisher Exact test or Chi-square test wherever appropriate. The participants’ predictors of positive perception towards the COVID-19 vaccination were determined using logistic regression analysis. In the first step, the relationship between independent variables and dependent variables was analyzed in the univariate analysis. All variables with *p* < 0.1 were included in the multivariable analysis in the first step. A *p* value ≤ 0.05 was considered to show a statistically significant test result.

## 3. Results

### 3.1. General Characteristics of Participants

One thousand nine participants completed the survey and were recruited for this study. Just over half of participants were male (52.6%) and the largest group of respondents were aged between 30 and 39 years (33.8%). These data are presented in Table 2. Chronic diseases were reported by only 14.2% of the study participants. Hypertension was the most common comorbidity (5%), followed by diabetes (3.6%), and chronic heart disease (2.7%). The majority of the study subjects did not use regular medications (81.1%). Most of the study participants had occupations (77.3%), and only 18.7% and 4% of respondents were unemployed or retired at the time of this study, respectively. Respondents displayed various educational levels, with the largest group having a bachelor’s degree (42%), followed by 38.4% of respondents holding secondary school degrees, and only 7.1 percent of the respondents indicating higher education. About one-fifth of respondents reported getting the flu shot in the past (21.2%), and prior COVID-19 infections were reported by 17.9% of the respondents. Nearly half of the participants learned about COVID-19 vaccines from social media (48.6%).

### 3.2. Knowledge, Attitude, and Perceptions towards COVID-19 Vaccines

In this study, 633 participants (representing 62.7% of the total study population) believed that the COVID-19 vaccination would produce an immune response against COVID-19 (Table 1). In addition, 80.8 percent of the respondents thought that everyone should get vaccinated to end the COVID-19 pandemic, and 83.8% of them were worried about family members becoming ill with COVID-19. Some patients (17.7%) had the false perception that people who have recovered from COVID-19 do not need to get vaccinated. About a quarter of participants also had another misconception: that the vaccine’s protection is achieved immediately after getting the COVID-19 vaccine. More importantly, 31.2% of the participants thought there was no need to take precautions such as social distancing, hand hygiene, and masking following COVID-19 vaccination.

As shown in Table 2, various factors have significant associations with a positive perception of COVID-19 vaccination. These include gender (*p* = 0.026), education level (*p* = 0.005), previous influenza vaccination history (*p* < 0.001), and personal history of COVID-19 infection (*p* = 0.014). In addition, more parameters were also compared between these two groups, including general knowledge and attitude statements on vaccines, such as “Everyone should get vaccinated to end the COVID-19 pandemic”, “people who have had COVID-19 and recovered have no need to get vaccinated”, and “children under the age of 15 should be vaccinated for COVID-19” (Table 3). Significantly more participants with a positive perception of COVID-19 vaccination believed that everyone should get vaccinated to end the COVID-19 pandemic (93.4% vs. 59.6%, *p* < 0.001), and children under the age of 15 should be vaccinated for COVID-19 (59.1% vs. 33.5%, *p* < 0.001). A majority of the participants with a positive perception of vaccines thought that the COVID-19 pandemic changed their approach to vaccination (54.7% vs. 27.4%, *p* < 0.001). Interestingly, more participants with a positive perception of the vaccine had the incorrect belief that people who have had COVID-19 and recovered have no need to get vaccinated (18.5% vs. 16.5%), and that the vaccine’s protection is achieved immediately after receiving the COVID-19 vaccine (33.2% vs. 7.2%, *p* < 0.001).

### 3.3. Predictors of Positive Perception towards COVID-19 Vaccines

We performed logistic regression to assess the association between baseline characteristics and positive perception regarding COVID-19 vaccination. Univariate logistic regression analysis showed that five factors, including younger age (<30 vs. ≥30, *p* = 0.005), gender (*p* = 0.026), educational background (*p* = 0.003), previous influenza vaccination history (*p* < 0.001), and personal COVID-19 history (*p* = 0.015) were associated with positive perception towards COVID-19 vaccines (Table 4). All variables with *p* < 0.1 in the univariate analysis were included in the multivariable analysis logistic regression analysis model. Eventually, the findings revealed that older people (≥30 vs. <30) were less likely to have a positive perception towards COVID-19 vaccines (OR = 0.70, 95% CI = 0.51–0.94). We also found participants with a previous history of influenza vaccines (OR = 2.01, 95% CI = 1.43–2.84) and a personal history of COVID-19 (OR = 1.58, 95% CI = 1.10–2.26) were more likely to have a positive perception regarding COVID-19 vaccines. Compared with participants with secondary school education or below, those with bachelor’s degrees or above were more likely to have a positive perception towards COVID-19 vaccination (OR = 1.47, 95% CI = 1.12–1.91).

## 4. Discussion

To the best of our knowledge, this is the first study that evaluated the overall perception of the Turkish population toward COVID-19 vaccination and the factors that affected their perception. Based on our findings, the participants’ perceptions were significantly more favorable among those who were younger, had higher education levels, had a previous influenza vaccination, and had a personal history of COVID-19 infection (Table 4). In our study, a total of 62.7% of the respondents who agreed to be vaccinated had a positive perception regarding the COVID-19 vaccination. Although the fact that our study was conducted in a population that had accepted vaccination against COVID-19 reduced its generalizability to the whole population, low positive perception was remarkable even in individuals who agreed to be vaccinated. Therefore, it was thought that the rate of positive perception in our study was higher than the expected rate of positive perception of the entire population.

COVID-19 has become a global health challenge, and the world is awaiting an effective vaccine to halt the ongoing pandemic [25]. Currently, multiple COVID-19 vaccines have been authorized for human use; many more remain in the early stages of development. However, vaccination may be delayed due to a negative attitude and perception, a lack of knowledge, and increased hesitancy. Vaccine hesitancy is routinely reported when a novel vaccine is introduced into the community [26]. Perceptions of vaccines and their safety are paramount factors that have been shown to influence vaccine hesitancy [27].

Moreover, the WHO deemed vaccine hesitancy one of the biggest ten threats to public health in 2019 [28]. Concerns have also been raised about the vaccine’s efficacy and safety against new COVID-19 variants [29]. Therefore, we must identify factors that may mediate hesitation about COVID-19 vaccines that could increase the acceptability of COVID-19 vaccines and population confidence.

A prior study revealed that approximately two-thirds of the respondents had positive perceptions of the presently available COVID-19 vaccines given in Saudi Arabia [30]. Similarly, in our cohort, a total of 62.7% of the respondents had a positive perception regarding COVID-19 vaccination in the survey, where everyone was vaccinated. Both pieces of data are consistent with a systematic review worldwide on willingness to receive vaccines, which found that about 66% had a positive attitude regarding COVID-19 vaccination [31]. Contrary to these studies, another study reported that over half (53.5%) of healthcare professionals had positive perceptions of the COVID-19 vaccination [32]. The discrepancy in the perception of the vaccine in the studies mentioned above could be attributed to the fact that they were conducted at different times since greater confidence has been developed after the massive vaccination campaigns over the past few months. Moreover, similar patterns were also observed during the flu pandemic when robust vaccine concerns were reported in the early pandemic; however, as the pandemic progressed, these concerns vanished, followed by an increase in the acceptance rate of vaccines by the entire population [33,34].

A growing number of studies have found that many factors can influence pandemic vaccine acceptance, such as past vaccination, trust in current health systems, vaccine recommendations from physicians, and perception of vaccine efficacy and safety [35,36]. The seasonal influenza vaccine is a well-known and widely used vaccine among people of all ages worldwide. Our study included this factor in determining the effect of prior vaccination history on participants’ perception of COVID-19 vaccination. Our findings revealed that the positive perception group reported a history of influenza vaccination more frequently than the negative perception group, at 25.3% and 14.4%, respectively (*p* < 0.001). Several other studies have found positive trends in the COVID-19 vaccine acceptance rate among people who have received influenza vaccination [37,38]. All the studies mentioned above and our study emphasize the positive role of vaccination history in informing constructive beliefs about vaccination, consistent with other earlier studies [39]. This trend among people could be linked to positive experiences with previous vaccinations.

One of the principal factors associated with attitude towards and perception of COVID-19 vaccination is educational background. The present study showed that people with a bachelor’s degree or higher education had a more positive perception than those who graduated from an elementary school or below. This is in line with prior research from the United States, which showed that individuals with higher education were considerably more likely to believe in the safety and efficacy of the vaccine and to receive a COVID-19 vaccine [40]. According to recently published studies in various countries worldwide, low education is related to a reduced willingness to receive the COVID-19 vaccine [41,42]. This is likely because people with a higher education level have better comprehension skills and easier access to information.

There are conflicting results in studies where gender influences perception of, attitude towards, and acceptance of the COVID-19 vaccine. We found that females had more positive views on vaccination than males. However, this relationship was found to be insignificant in multivariate analysis. This finding aligns with another study in Bangladesh that did not show significant gender differences in perception toward COVID-19 vaccination [43]. However, most studies on gender-based predisposition to COVID-19 vaccine acceptance indicate that women tend to be more concerned about vaccine safety and more vaccine-hesitant [26,44]. It was also shown that men’s knowledge scores about COVID-19 were marginally higher than those of females [45]. Positive perceptions about the COVID-19 vaccine decreased with age in our study. The positive perception was highest among subjects aged 18–29 years (69%) and lowest among those aged 40–49 years (58%). These findings are similar to the results of a study conducted among adults in the United States, which demonstrated that participants aged 18–29 years exhibited higher acceptance (71%) than participants aged 50–64 years (64%) [46]. On the other hand, other studies have found that acceptance increases with age [47,48]. Regional differences in population perceptions and beliefs regarding vaccination, which differ by age groups, could explain such conflicting findings.

Although the significant developments and advances regarding COVID-19 vaccination that have been made with the help of the collaboration of the scientific community are astonishing, the issue is susceptible to contamination and weak scientific evidence, possibly due to research exceptionalism [49]. An exciting study reported a significant correlation between organizations on social media and public skepticism about vaccine safety [50]. There is misinformation and disinformation on social media and television [51], leading to incorrect beliefs about COVID-19 vaccines. A recent study among participants who were moderately hesitant about COVID-19 vaccines found that after providing the correct information about the efficacy and safety of COVID-19 vaccines, their intention to receive the vaccines increased compared with those who did not receive the same information [52]. Given that social media is the primary source of information on COVID-19 vaccines in our study population, tackling misinformation and disinformation on social media is crucial to reversing the growth in vaccine hesitancy around the world. Moreover, the negative perception towards vaccination was lower in participants who healthcare professionals informed about the source of information about the vaccine (*p* = 0.041). This finding emphasizes the importance of the physician’s role in vaccination, and physicians should improve their knowledge and confidence to make strong vaccine recommendations to their patients.

Health policies aiming to vaccinate the highest possible proportion of the population will be crucial to mitigating the impacts of the pandemic. This study defined several factors which mediated the positive perception regarding the COVID-19 vaccination, such as higher education level, having received the influenza vaccine in the past, and having a history of COVID-19 infection. The findings of this study are important for health policy makers and healthcare providers and can help better guide COVID-19 vaccine compliance. Specifically, efforts should be made to target males, the low-education population, and those who did not vaccinate themselves against influenza in the previous season. In addition, public health intervention programs should put more focus on increasing the perception of the disease’s severity due to the higher positive perception of those with a history of COVID-19 infection. Finally, more resources should be allocated by the Ministry of Health to inform the public about COVID-19 vaccination campaigns and to develop appropriate immunization education programs to guide the management of possible future pandemics. However, several limitations affect the proper interpretation of the study findings. Firstly, we could not use a previously validated questionnaire, as there was no valid questionnaire in the Turkish population throughout the course of the study.

Another critical point is that the study was cross-sectional, and the temporal relationship is unknown; therefore, the causality cannot be attributed to the findings in the regression models. Another limitation is the paucity of data regarding healthcare workers and older adults. Despite the high number of participants, the fact that the study includes data from a single center prevents our study from being generalized. In these settings, the findings might not be enough to be representative of the entire population.

## 5. Conclusions

The COVID-19 pandemic continues to wreak havoc on lives and livelihoods worldwide, and the COVID-19 vaccine represents a possible glimmer of hope for the future. This study provides insight into the Turkish population’s knowledge, attitudes, and perceptions regarding COVID-19 vaccines. Overall, most of our study participants’ positive perceptions of the efficacy of current COVID-19 vaccines is consistent with many reports of attitude and perception around the world. A more nuanced and updated understanding of vaccines’ perception and determinants is urgently needed to tailor public health messages accordingly.

## Figures and Tables

**Table 1 vaccines-10-00278-t001:** Knowledge and practice of study participants toward COVID-19 vaccine in Turkey.

Statement		Total *n* (%)
Do you believe that the vaccines produce an immune response against COVID-19?	Agree	633 (62.7)
	Disagree	376 (37.3)
Everyone should get vaccinated to end the COVID-19 pandemic.	Agree	815 (80.8)
	Undecided	132 (13.1)
	Disagree	62 (6.1)
Although we cannot vaccinate everybody, the pandemic will end soon.	Agree	322 (31.9)
	Undecided	305 (30.2)
	Disagree	382 (37.9)
Children under the age of 15 should be vaccinated for COVID-19.	Agree	500 (49.6)
	Undecided	365 (36.2)
	Disagree	144 (14.3)
People who have had COVID-19 and recovered have no need to get vaccinated.	Agree	179 (17.7)
	Undecided	318 (31.5)
	Disagree	512 (50.7)
The vaccine’s protection is achieved immediately after getting the COVID-19 vaccine.	Agree	237 (23.5)
	Undecided	479 (47.5)
	Disagree	293 (29.0)
I can stop practicing precautions such as masking, social distancing, and hand hygiene after receiving the COVID-19 vaccine.	Agree	185 (18.3)
	Undecided	136 (13.5)
	Disagree	688 (68.2)
The COVID-19 pandemic changed my approach to vaccination.	Agree	449 (44.5)
	Undecided	283 (28.0)
	Disagree	277 (27.5)
I am worried about family members becoming ill with COVID-19	Agree	846 (83.8)
	Undecided	77 (7.6)
	Disagree	86 (8.5)

**Table 2 vaccines-10-00278-t002:** Baseline characteristics and comparison of adequate perceptions (*n* = 1009), * *p* < 0.05.

Variables	*n* (%)	Negative Perception; *n* (%)	Positive Perception; *n* (%)	*p* Value
Age				
18–29 years	290 (28.7)	89 (23.7)	201 (31.8)	0.086
30–39 years	341 (33.8)	132 (35.1)	209 (33.0)	
40–49 years	308 (30.5)	127 (33.8)	181 (28.6)	
50–59 years	61(6.0)	25 (6.6)	36 (5.7)	
60 years and above	9 (0.9)	3 (0.8)	6 (0.9)	
Gender				
Female	478 (47.4)	161 (42.8)	317 (50.1)	0.026 *
Male	531 (52.6)	215 (57.2)	316 (49.9)	
Chronic Disease				
No	866 (85.8)	332 (88.3)	534 (84.4)	0.083
Yes	143 (14.2)	44 (11.7)	99 (15.6)	
Hypertension	50 (5.0)	11 (2.9)	39 (6.2)	
Asthma/COPD	6 (0.6)	3 (0.8)	3 (0.5)	
Diabetes	36 (3.6)	14 (3.7)	22 (3.5)	
Chronic Heart Disease	27 (2.7)	9 (2.4)	18 (2.8)	
Cancer	4 (0.4)	2 (0.5)	2 (0.3)	
Chronic Kidney Disease	11 (1.1)	2 (0.5)	9 (1.4)	
Other	33 (3.3)	12 (3.2)	21 (3.3)	
Previous medication use				
Yes	191 (8.9)	72 (19.1)	119 (18.8)	0.891
No	818 (81.1)	304 (80.9)	514 (81.2)	
Educational level				
No education	14 (1.4)	8 (2.1)	6 (0.9)	0.005 *
Primary school	112 (11.1)	50 (13.3)	62 (9.8)	
Secondary school	387 (38.4)	156 (41.5)	231 (36.5)	
Bachelor degree	424 (42.0)	146 (38.8)	278 (43.9)	
Higher Education	72 (7.1)	16 (4.3)	56 (8.8)	
Occupational Status				
Employed	780 (77.3)	290 (77.1)	490 (77.4)	0.763
Non-employed	189 (18.7)	73 (19.4)	116 (18.3)	
Retired	40 (4.0)	13 (3.5)	27 (43)	
Influenza shot in the past				
Yes	214 (21.2)	54 (14.4)	160 (25.3)	<0.001 *
No	795 (78.8)	322 (85.6)	473 (74.7)	
Having a child				
Yes	545 (54)	200 (46.8)	345 (54.5)	0.686
No	464 (46)	176 (46.8)	288 (45.5)	
My child received all the recommended vaccines				
Yes	499 (91.4)	172 (85.6)	327 (94.8)	<0.001 *
No	47 (8.6)	29 (7.7)	18 (5.2)	
Personal history of COVID-19				
Yes	181 (17.9)	53 (14.1)	128 (20.2)	0.014 *
No	828 (82.1)	323 (85.9)	505 (79.8)	
Sources of COVID-19 vaccine-related information (multiple choices)				
Recommendation from health authorities	200 (19.8)	62 (16.5)	138 (21.8)	0.041 *
Social media	490 (48.6)	172 (45.7)	318(50.2)	0.167
Friends	97 (9.6)	44 (11.7)	53 (8.4)	0.083
Recommendation from my relatives who are HCWs	147 (14.6)	53 (14.1)	94 (14.8)	0.743
Other sources	251 (24.9)	118 (31.4)	133 (21.0)	<0.001 *

**Table 3 vaccines-10-00278-t003:** Comparison knowledge and practice of study participants toward COVID-19 vaccine according to perception. * *p* < 0.05.

Statement		Negative Perception; *n* (%)	Positive Perception; *n* (%)	*p* Value
Everyone should get vaccinated to end the COVID-19 pandemic.	Agree	224 (59.6)	591(93.4)	<0.001 *
	Undecided	101 (26.9)	31 (4.9)	
	Disagree	51 (13.6)	11 (1.7)	
Although we cannot vaccinate everybody, the pandemic will end soon.	Agree	101 (26.9)	221 (34.9)	0.005 *
	Undecided	134 (35.6)	171 (27.0)	
	Disagree	141 (37.5)	241 (38.1)	
Children under the age of 15 should be vaccinated for COVID-19.	Agree	126 (33.5)	374 (59.1)	<0.001 *
	Undecided	164 (43.6)	201 (31.8)	
	Disagree	86 (22.9)	58 (9.2)	
People who have had COVID-19 and recovered have no need to get vaccinated.	Agree	62 (16.5)	117 (18.5)	<0.001 *
	Undecided	153 (40.7)	165 (26.1)	
	Disagree	161 (42.8)	351 (55.5)	
The vaccine’s protection is achieved immediately after getting the COVID-19 vaccine.	Agree	27 (7.2)	210 (33.2)	<0.001 *
	Undecided	220 (58.5)	259 (40.9)	
	Disagree	129 (34.3)	164 (25.9)	
I can stop practicing precautions such as masking, social distancing, and hand hygiene after receiving the COVID-19 vaccine.	Agree	46 (12.2)	139 (22.0)	<0.001 *
	Undecided	70 (18.6)	136 (13.5)	
	Disagree	260 (69.1)	428 (67.6)	
The COVID-19 pandemic changed my approach to vaccination.	Agree	103 (27.4)	346 (54.7)	<0.001 *
	Undecided	140 (37.2)	143 (22.6)	
	Disagree	133 (35.4)	144 (22.7)	
I am worried about family members becoming ill with COVID-19	Agree	256 (68.1)	590 (93.2)	<0.001 *
	Undecided	51 (13.6)	26 (4.1)	
	Disagree	69 (18.4)	17 (2.7)	

**Table 4 vaccines-10-00278-t004:** Univariable and multivariable logistic regression analysis of influencing factors for positive perception to COVID-19 vaccination. * *p* < 0.05.

	Univariate Analysis			Multivariate Analysis		
Predictive Variable	OR	95% CI	*p* Value	OR	95% CI	*p* Value
Age (≥ 30 vs. <30)	0.66	0.49–0.88	0.005 *	0.70	0.51–0.94	0.020 *
Gender (female vs. male)	1.34	1.03–1.73	0.026 *	1.29	0.99–1.68	0.053
Chronic disease (yes vs. no)	1.39	0.95–2.04	0.084	1.38	0.94–2.04	0.100
Previous medication use (yes vs. no)	0.97	0.70–1.35	0.891			
Education level (≥Bachelor degree vs. ≤Secondary school)	1.47	1.14–1.90	0.003 *	1.47	1.12–1.91	0.004 *
Influenza shot in the past year (yes vs. no)	2.01	1.43–2.83	<0.001 *	2.01	1.43–2.84	<0.001 *
Personal history of Covid-19 (yes vs. no)	1.54	1.08–2.19	0.015 *	1.58	1.10–2.26	0.012 *

## Data Availability

The data that support the findings of this study are available on request from the corresponding author. The data is not publicly available due to privacy or ethical restrictions. Meliha Cagla Sonmezer; ORCID ID: 0000-0001-6529-5282.

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
