# Peer review of "Knowledge, Attitudes, and Perception towards COVID-19 Vaccination among the Adult Population: A Cross-Sectional Study in Turkey"

_vaccines, 2022, doi:10.3390/vaccines10020278_

Round 1
Reviewer 1 Report
This article presents a study on perceptions, attitudes, and knowledge towards anti-COVID vaccination in a population sample from Turkey. The topic of the study is very current since about a year and there are many similar works in the literature; however, the authors claim that this is the first study of this kind conducted in Turkey, which provides new information on the topic. The design of the study echoes that of other existing works on the subject but is formally correct. The introduction is comprehensive and informative. In the materials and methods section, the exposition and information needs to be improved: specifically, the part concerning the last section of the questionnaire needs to be clarified better, e.g. how were the 8 questions chosen? On what basis the authors decided to divide the perception towards vaccination according to the answer given to only one question? The authors should improve the scientific soundness of this section. The results section is fairly well organized but there are some parts that are redundant (e.g. first paragraph) or need to be moved to methods (e.g. line 156-160). The results are discussed satisfactorily and an adequate number of references are cited. In the text there are several typos to correct and the English language to check.
Author Response
Thank you for the constructive criticism and insightful comments. We diligently worked to address each of these comments constructively. This manuscript has been read and approved by all the authors. Our responses to the reviewer's comments are given point by point.

Reviewer 2 Report
The manuscripts addressed an important public health problem during the current pandemic era, which is the vaccine hesitancy and the factors affecting the vaccine acceptance in this population.
Generally, the manuscript is clear, well-structured and addressed the objectives of the study. However, there was a lack in consistency in the use of the main outcome variables “knowledge, attitudes, practices, perceptions” throughout the abstract, introduction particularly the aim, the results and the discussion.
I think it is quite difficult to generalize the results of this study on the whole Turkish population that had been mentioned several times throughout the study for many factors, some are mentioned in the limitations of the study and others are not mentioned, like the source of the studied group, the sampling technique used for their inclusion in the study and the limited inclusion of the people aged 65+ since they had already received the vaccine. I think, inclusion of the people attending the assigned hospital to receive their first dose of the vaccine as a group to assess the perception of the general public toward the vaccine can be biased since they are already attending to have the vaccine, considering that the vaccination is not compulsory. Further clarification of this point is very crucial.
Although the authors mentioned that they did not find a validated questionnaire to use in this study, but they did not mention how the data collection tool was developed and if they test its validity. The results can be reproduced if these points in addition to clarifying the source of the study population and the sampling used are clarified.
The main factors that drive the perception in this study were the educational level, having influenza shot and the history of contracting COVID-19, and I can see that all of them are non-modifiable for the time being. How the policymakers will get use of the findings of this study to shape their strategies towards minimizing vaccine hesitancy and increase vaccine absorptivity?
Lines 20,21, the word general is not convenient
Line 23, Most is not the suitable word to show slight difference in gender distribution
Line 24, Range is not the suitable word to demonstrate the highest proportion
Same these words were also used in the methods and the results. Please consider revising them
Lines 32,33, it was supposed that this manuscript will help addressing the perceptions and the factors affecting it.
Line 43, I think it is better to update the global figure at least to the beginning of this year, Jan,1,2022
Line 44, "since" is not the right word
Line, 49, "currently", I think this statement is old and the reference lacks the date. Please revise and use a more recent WHO SAGE document
I think it is important to highlight the vaccine efficacy across the different variants, which is quite variable.
Line 63, 25%?
Line 85, "no previous studies", I think there are some studies which I found on google.com addressing KAP about COVID-19 vaccine in Turkey in different settings.
Line 93, 94 further elaboration on the study participants, selection criteria is needed
Table 1 is inserted in the methods?
Line 151, instead of minority please mention the exact proportion.
Table 2, for the chronic diseases, I do not know how the chi square was calculated here
Just for a curiosity, since the bivariate analysis was done and the p values were identified, why you did not go multiple regression analysis?
Author Response

(The authors gave the same response as above.)

Round 2
Reviewer 2 Report
Thanks to the authors for addressing my comments.